# Design of U-Shaped Frequency Tunable Microwave Filters in MEMS Technology [note 1]

**DOI:** 10.3390/s23010466

**Published:** 2023-01-01

**Authors:** Flavio Giacomozzi, Emanuela Proietti, Giovanni Capoccia, Giovanni Maria Sardi, Giancarlo Bartolucci, Jacopo Iannacci, Girolamo Tagliapietra, Benno Margesin, Romolo Marcelli

**Affiliations:** 1Fondazione Bruno Kessler, 38123 Trento, Italy; 2CNR-IMM, 00133 Roma, Italy; 3Electronic Engineering Department, University of Roma “Tor Vergata”, 00133 Roma, Italy

**Keywords:** RF MEMS, microwave resonators, frequency tunability

## Abstract

U-shaped microwave resonators implemented by RF MEMS switches can be considered the result of a novel design approach for obtaining small-footprint tunable resonators, owing to the bent shape of the resonator and the microsystem solution for changing the frequency of resonance. In this paper, we discuss the design approach for potential configurations of U-shaped structures combined with ohmic RF MEMS switches. Owing to their prospective application in RADAR and satellite systems, the devices were assessed for K-Band operation, specifically for 15 GHz, 20 GHz, and 26 GHz. The ON-OFF states determined by an electrostatic actuation of metal beams composing the RF MEMS ohmic switches allow for selecting different path lengths corresponding to different frequencies. In this contribution, initial configurations were designed and manufactured as a proof-of-concept. The advantages and critical aspects of the designs are discussed in detail.

## 1. Introduction

The use of RF MEMS (radio frequency micro-electro-mechanical system) switches to provide signal routing and frequency tunability of microwave and millimeter wave components and antennas has been studied over the last two decades. Reports on state-of-the-art designs and past literature can be found in [1,2,3,4,5,6]. This topic relates to the general need to study and apply reconfigurable components and sub-systems, as outlined in [7].

The optimization of the RF MEMS response in terms of mechanical reliability and charge injection (mainly in the dielectric pads used for the electrostatic actuation) was obtained in several cases, together with the well-known advantages of low insertion losses and an overall passive environment for high-frequency signal processing [8,9,10,11,12,13].

In this work, we shall perform a theoretical study on compact, coplanar waveguide (CPW) configurations of U-shaped resonators to be used for narrow-range digital frequency tunability. U-resonators have already been studied for realizing microstrip-based hairpin filters to obtain a more compact layout for coupled line filters [14]. An additional utilization is in periodic structures designed for high-frequency meta-surfaces [15].

The most critical challenge in using a U-shape is to optimize the dimensions for hosting the resonator and the switch at the same time in a small area, accounting for the necessary footprint of the MEMS. The starting point for a suitable design is the requirement to have good expected performances for a simple U-shaped resonator not implemented for frequency tuning, considered as a reference device at a fixed frequency. This consideration implies the electrical matching of the naked resonator, including a good coupling with the central conductor of a CPW.

The goal for the design was to inspect resonators for the K-Band, with the possibility of including two switches for devices working around 20 GHz and one only around 26 GHz. Further results have been obtained with a resonator implemented with a single cantilever working at 15 GHz.

The limitation in using two resonators or just one depends on the actual size of the switch which includes a metal beam with a double-clamped structure and feeding lines and slots where the MEMS must be hosted. The frequencies have been chosen following the specific requirements for TX/RX modules to be used in satellite applications.

Band-stop resonators have been manufactured and tested with good performances for naked configurations (without switches). Furthermore, the design approach for implementing them with RF MEMS switches has been validated using 2D simulations.

In this work, we extended the results presented in [16] by comparing resonators working at different frequencies and including additional considerations about the coupling of U-structures in CPW configuration, with care for the advantages of using symmetric structures.

Despite there being several contributions in the literature about U-shaped resonators, no efforts are available, to the authors’ knowledge, examining CPW-designed U-resonators implemented in MEMS technology. The novelty of our contribution is in the general theoretical and experimental analysis that has been performed. This paper aimed to study: (i) the possibility of including micromechanical devices in the resonator structure with consideration for the RF MEMS footprint by comparing double-clamped and cantilever switches, and (ii) the importance of symmetrical geometries to improve the electrical coupling of the resonator to the CPW feeding.

## 2. Design and Simulations

U-shaped resonators are simple and compact resonating structures that can be coupled in microstrip or coplanar configurations. They are also suitable for realizing filters based on mutual coupling or even absorbing metamaterial structures up to THz frequencies. The naked band-stop resonators (without switches) were designed accounting for the U-shape’s full length, predicting resonance frequencies for a nominal operation around 26 GHz, 20 GHz, and 15 GHz ca. A coplanar waveguide (CPW) configuration was chosen. The CPW solution is helpful for taking advantage of the in-plane confinement of the electric field and the negligible interference between adjacent planes in case of utilization of the resonator in a stack. RF MEMS switches were initially considered in a double-clamped structure. This configuration is mechanically and electrically favored because the lateral ground planes where the MEMS bridge is anchored and grounded can be easily defined. The cantilever solution was also studied because it is a less invasive structure.

Simulations were performed using the Microwave Office software based on the AXIEM environment and released by AWR-CADENCE [17]. This initial device structure allowed us to conduct a preliminary evaluation of the influence on the electromagnetic response of the resonator in a 2.5D simulation. The simulated configurations and the expected performances for the naked band-stop U-shaped resonators are shown in Figure 1 and Figure 2, respectively. At the same time, Table 1 gives the detailed dimensions of the resonators.

The exact die total area is 3160 × 3210 μm^2^ for both the 26 GHz and 20 GHz resonators. In addition, one-third of the device is coupled with the central conductor of the CPW, and the vertical arms are characterized by lengths A = 980 µm and 725 µm. The width *W* of the coplanar lines is 90 µm, and the gap to the lateral ground planes *G* is 60 µm to provide a 50 ohm matching for the device input/output; the primary dimensions of the resonators are L_1_ = 810 µm and L_2_ = 1160 µm. Therefore, the arm widths are A_1_ = 980 µm and A_2_ = 725 µm. In this stage, we considered a square footprint for both resonators. In the case of the 15 GHz resonator, we maintained the exact value of the 20 GHz resonator for the length of the vertical arm, i.e., A_3_ = 980 µm = A_1_, and L_3_ = 1950 µm.

## 3. Technology

The manufacturing process to obtain the naked U-shaped resonators was performed using the technological facility of the FBK partner of the project.

The technology for the reference devices without switches is based on one-mask-only photolithography. An eight-mask surface micromachining process is necessary to fabricate RF MEMS switches [18].

Figure 3 shows a schematic cross-section indicating the structure of an electrostatically actuated ohmic switch. The substrates are 6-inch diameter high resistivity silicon wafers with 1 micron of thermal oxide. The resonators’ CPW transmission lines and grounds structure were produced using thick electroplated gold. A first short run was used for the naked devices (CPW structures only with no movable switches), which were grown by electroplating 5 μm of gold inside the patterned thick photoresist. In the complete process, two layers of electroplated gold must be used to obtain the switch’s suspended membrane. The deformable parts were obtained with the thinner one, 1.8 µm thick. The second one, 3.5 µm thick, must be superimposed to increase the rigidity of the membranes’ selected parts and to obtain the CPW structures. The DC electrodes are used as actuation pads of the electrostatic switches, and the bias lines and contact bumps are defined using high-resistivity polysilicon. Under the suspended switch membranes, the underpass signal line was realized using a 630 nm Ti-TiN-Al multilayer, and 150 nm of gold was evaporated over the contact areas to obtain low-resistance contacts. The air gap between the actuation electrodes and the movable membrane is obtained by a 3 µm thick photoresist sacrificial layer removed by oxygen plasma at the end of the process.

The fabrication process started with the oxidation of the silicon wafers. A 630-nm thick polysilicon layer was deposited by low-pressure chemical vapor deposition (LPCVD), lightly doped by B ion implantation and defined by lithography and dry etching. An insolation layer of 300 nm silicon dioxide was deposited by the LPCVD technique, and the contact holes were defined inside. A multilayer consisting of a Ti/TiN diffusion barrier and Al1%Si was sputtered over the wafers and patterned. The 100-nm thick SiO_2_ was then deposited over it by PECVD, and contact holes were realized. A 5 nm Cr–150 nm Au thin film was then evaporated by an electron beam gun, patterned, and wet etched. A 3-micron thick sacrificial photoresist (spacer) was spun and lithographically defined to realize suspended and movable structures. A 2.5 nm Cr and 25 nm Au seed layer was then evaporated all over the wafer to realize an electrically conductive surface for the gold electroplating. A thick AZ 4562 resist mold was defined over the wafer, and the 1.8-μm bridge gold layer was electroplated inside the cavities. After the resist removal, a second AZ 4562 mold was defined for the selective electroplating of the thicker (3 µm) CPW gold layer. After the removal of the resist and the Cr-Au seed layer, the sacrificial resist spacer was burned by oxygen plasma to release the suspended movable structures.

Test devices were manufactured for an early-stage control of the correctness of the design, comparing naked resonators with ideal devices implemented with collapsed beams or open circuits to emulate closed and open switches. The collapsed beams were obtained without using a sacrificial layer, which is normally deposited and successively removed to obtain the suspended membrane; in this case, we say that the device is “technologically actuated”. The open circuit ideal situation is vice versa, characterized by having no metal beam on the surface, leaving a capacitive gap along the central conductor of the CPW.

## 4. Experimental Results and Discussion

The devices were measured on-wafer using a two-port network analyzer system calibrated with standard lines manufactured directly on the same substrate, following the TRL (Thru-Reflect-Line) calibration procedure. The results obtained for naked and technologically actuated or open circuit U-shaped resonators (ideal devices) are shown and discussed in the following sections and figures.

### 4.1. Naked Devices (U-Shaped Resonators without Switches)

The ideal devices (naked configurations) with no RF MEMS switches were characterized on-wafer using a two-port vector network analyzer. The resonators and their electrical performances are shown in Figure 4 and Figure 5.

The shifts experienced in the frequency of resonance are probably due to software limitations in simulating coupled lines and the inexact knowledge of material parameters such as the dielectric constant and the silicon dioxide thickness. A contribution to the resonance frequency shift is probably due to the non-ideal shape of the coupled lines. They are subjected to photolithography and electroplating. The last process is necessary to minimize the skin depth effect, and increasing the gold thickness modifies the vertical shape of the edge of the line, involving some irregularities in the slots. Finally, it modifies the equivalent capacitance, contributing to a change in the coupling and a shift in the resonance frequency.

### 4.2. Technologically Actuated, Semi-Ideal Devices with Clamped-Clamped Beams

Devices with technologically actuated switches were manufactured as intermediate, semi-ideal test structures. This condition does not include the sacrificial layer below the double-clamped structure that is added during manufacturing. The 20 GHz resonators have the right arm or the left arm of the switch opened to emulate the capacitance when the metal beam is up. At the same time, the other one is technologically actuated. The 26 GHz resonator has one switch only because of the footprint size limitation previously discussed.

The manufactured devices are shown in Figure 6.

The measured performances of both intermediate configurations are plotted in Figure 7. The figure represents the two possible states for the 20 GHz operation and the response of the 26 GHz when the switch is down. The presence of the switches alters the desired frequency of resonance in a significant way. For the 20 GHz resonator, this is an expected result. Furthermore, additional elements for frequency tuning lead to a central frequency variation of the band-stop device. In the case of the 26 GHz device, this effect is enhanced. The reason is that the switch’s size is always the same but placed on a smaller resonator, thus interfering with its electromagnetic response in a more significant way, with a 2 GHz shift down for the naked resonator.

### 4.3. Technologically Actuated, Semi-Ideal Device with Cantilever

The 15 GHz resonator was studied by introducing a switch made by a cantilever. The advantages of this configuration are a single ohmic contact, which decreases the insertion loss by a factor of two, and a smaller footprint of the RF MEMS device. On the other hand, this structure must be manufactured more carefully in real cases because the residual mechanical stress on the released metal beam could contribute to unwanted upward or downward bending of the cantilever.

The cantilever-based switches adopt for the actuation a mechanism like that of a double-clamped beam used for the previous breadboards. In this case, the DC electrostatic actuation is provided by an actuation pad manufactured below the cantilever, fed by an external pad connected using a polysilicon highly resistive line. The actuation occurs around the middle of the beam. This solution allows the collapse of the end of the cantilever, resulting in a metal-to-metal contact located along the central conductor of the CPW. The same principle of operation is followed with the double-clamped structure but includes two metal contacts instead of one only.

The studied structure is represented in Figure 8, where the single resonator is shown together with the one loaded by the RF MEMS switch and a detailed view of the cantilever.

The electrical contact is provided by closing (collapsing) the cantilever RF MEMS switch with electrostatic actuation. The cantilever, analogously to the double-clamped structure, is made with holes to efficiently remove the sacrificial layer below. The presence of the holes also decreases the beam’s mechanical stress and stiffness and, consequently, the device’s actuation voltage.

The simulation of the device, as shown in Figure 8, was performed with either the absence (naked device) or presence of the RF MEMS switch. The results of the simulation for both situations are shown in Figure 9 and demonstrate the effect of the additional device on the frequency of resonance and the electrical matching.

The experimental results on the 15 GHz device are shown in Figure 10. A comparison is performed among: (i) the U-resonator without the switch, (ii) the one loaded by the switch technologically actuated, and (iii) the resonator with a capacitive gap to emulate the up-state of the cantilever. It is evident that despite the non-perfect electrical matching, the switch in the actuated state marginally affects the electrical performance of the resonator. The frequency of resonance is increased by opening the circuit along the CPW feed line. This occurs because the full length of the resonator is decreased and the corresponding intrinsic wavelengths are also decreased.

### 4.4. Performance Implementation of the U-Shaped Devices by Symmetric Configurations

The CPW configurations used to develop the studied resonators are characterized by a symmetric arrangement of the ground planes with respect to the central conductor. The measured performance is sufficient for many applications, but an enhanced performance of the stopband filter can be obtained by introducing a symmetric structure. For this reason, an additional prototype was produced to assess a symmetric geometry around the central conductor of the CPW. In particular, the 20 GHz resonator was modified in a mirror-like configuration, doubling the U-resonators, as shown in Figure 11.

This solution is still acceptable from the space occupancy point of view of the structure, and it increases the coupling efficiency, leading to a deeper notch of the band-stop resonator. It must be remembered that the designed resonators are not thought of as notch filters but as resonating structures with a fractional band in the order of 5%. The mirror implementation is considered the natural evolution of the initially proposed devices and does not result in any unwanted increase in the device footprint. The lateral ground plane is typically at least five times larger than the CPW, and it can easily host a mirror resonator, with possible additional switches to improve the tunability.

To complete the test on the symmetric configurations, the same structure for the 26 GHz operation was also evaluated, and the theoretical and experimental results are given in Figure 12.

The results in Figure 12 confirm that mirroring the structure is essential to having more pronounced peaks at resonance, thus obtaining a better electrical matching. The situation is explained in Figure 13, where the simulation of the single resonator and that of the mirrored one are plotted together.

On the other hand, it is more difficult to precisely filter the frequencies closer to the millimeter wavelength range at the chosen frequency without interferences due to additional modes excited by increasing the number of discontinuities involved in the complete device.

A structure implementation is needed for this specific configuration or for possible use at higher frequencies to match exactly the wanted resonance frequency. It is worth noting that this effect could be less critical at lower frequencies, even if the operating frequency is not much smaller, as shown in Figure 14.

An important consideration concerning the symmetric U-shaped resonators is whether they can host switches on both sides of the structure, increasing the number of possible frequencies to be tuned.

To complete the picture of the resonator responses, we also simulated the 15 GHz structure by mirroring the resonator with respect to the central conductor of the CPW. The situation is more intriguing in this case because an additional mode is excited, and a dual-frequency operation is obtained. The situation is described in Figure 15, comparing, as before, the single and the mirrored resonator simulations.

The result implies a general consideration about this geometry. A dual-frequency operation is sometimes required in system applications, but the initial expectation was to have a single-mode resonator. Therefore, a theoretical and experimental investigation is required for frequencies lower than the K-Band. This could be helpful to understand better if a multi-resonant response is generally expected or if the main mode has been subjected to a break of the degeneracy, creating two possible resonance modes originally corresponding to the same frequency for the studied resonator. A perturbation of a resonating system always causes a change with respect to a ground situation. One of the possible outputs is to excite two frequencies around the original one. In this specific case, it must be understood how the coupling mechanism is responsible for that and if specific frequencies related to the geometry can be considered to activate this mechanism.

## 5. Conclusions

This paper proposes that U-shaped band-stop resonators can be developed in MEMS technology as a compact solution for narrow-band microwave tuning. A step-by-step analysis was performed ranging from the naked devices to the ideal resonators with RF MEMS switches technologically actuated or emulated by open gaps for the up-state of the beam. K-band frequencies were assessed for future implementation in devices for satellite communications. The advantages and criticalities in realizing the devices were discussed considering the design and experimental findings, as well as improvements using symmetric configurations. 

## Figures and Tables

**Figure 1 sensors-23-00466-f001:**
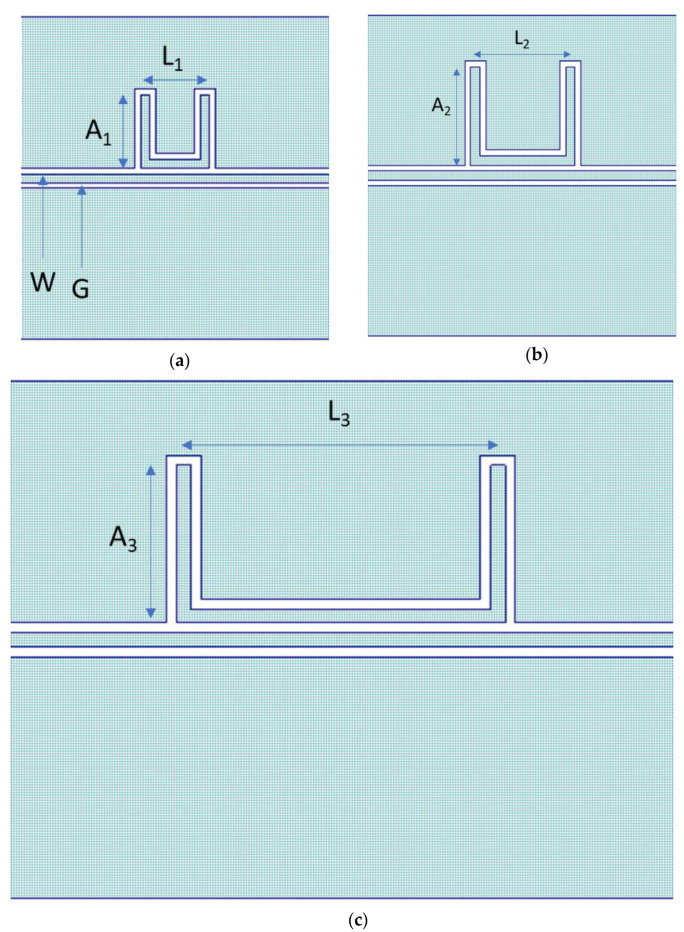
Design of (**a**) 26 GHz, (**b**) 20 GHz, and (**c**) 15 GHz resonators.

**Figure 2 sensors-23-00466-f002:**
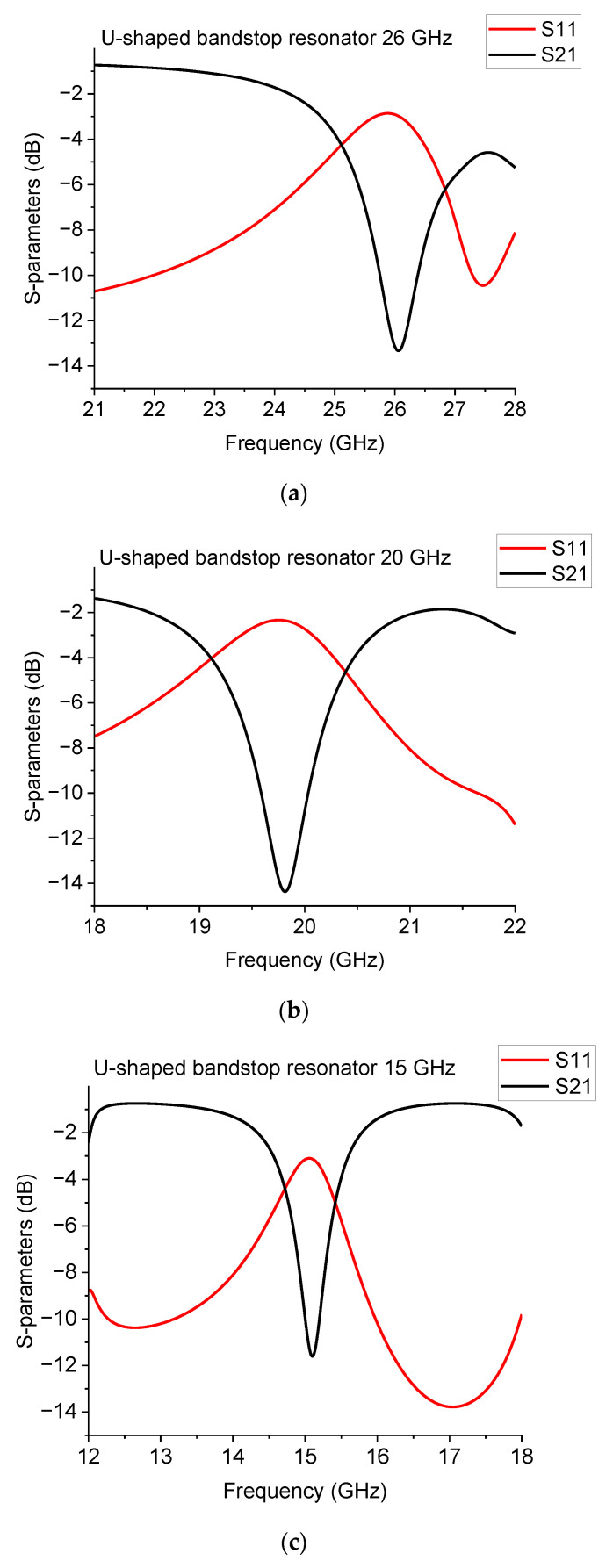
Simulated response for the U-shaped resonators, working approximately around (**a**) 26 GHz, (**b**) 20 GHz, and (**c**) 15 GHz. S11 is the reflection parameter, while S21 is the transmission parameter.

**Figure 3 sensors-23-00466-f003:**
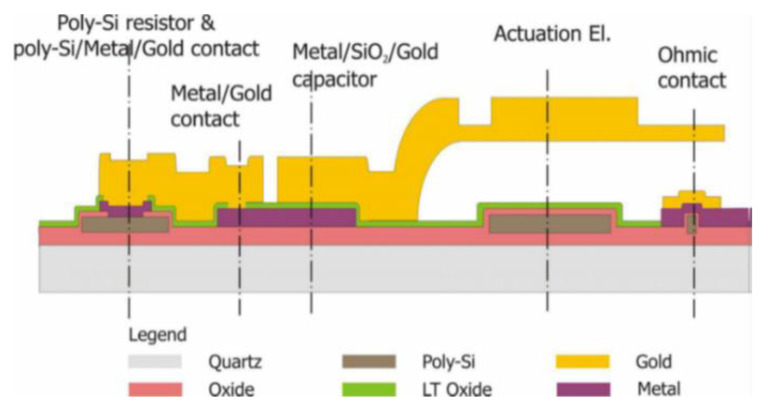
Cross-section view of the FBK-RF-MEMS fabrication process for ohmic-contact switches.

**Figure 4 sensors-23-00466-f004:**
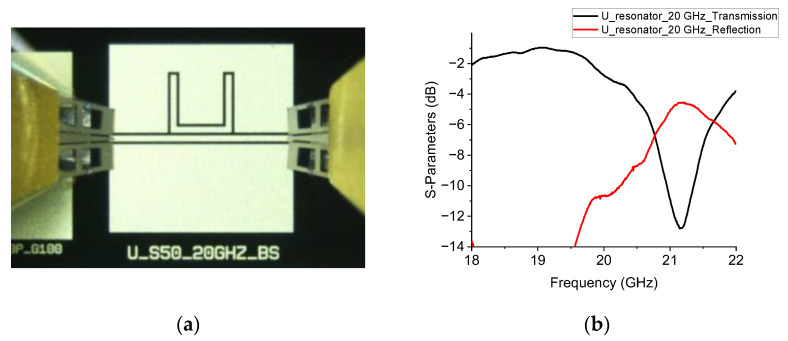
Naked configuration device (**a**) and measured performance (**b**) for the 20 GHz U-shaped resonator.

**Figure 5 sensors-23-00466-f005:**
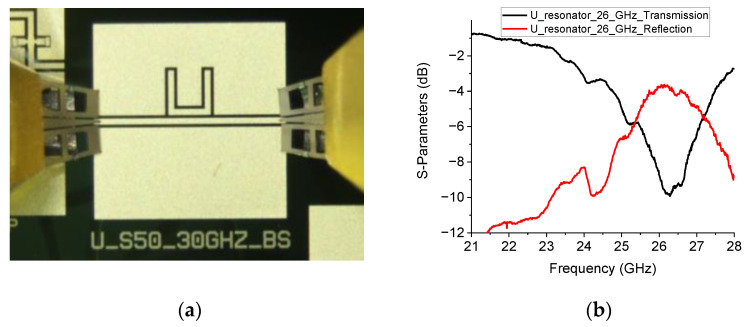
Naked configuration shape (**a**) and measured response (**b**) for the 26 GHz device.

**Figure 6 sensors-23-00466-f006:**
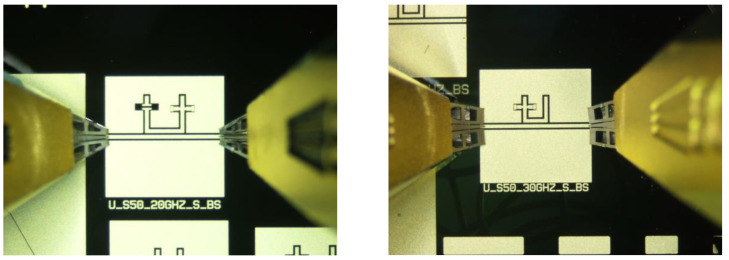
On the left is the 20 GHz resonator and, on the right, the 26 GHz one, where the switches are technologically actuated (completely collapsed beam) or not technologically actuated (beam removed).

**Figure 7 sensors-23-00466-f007:**
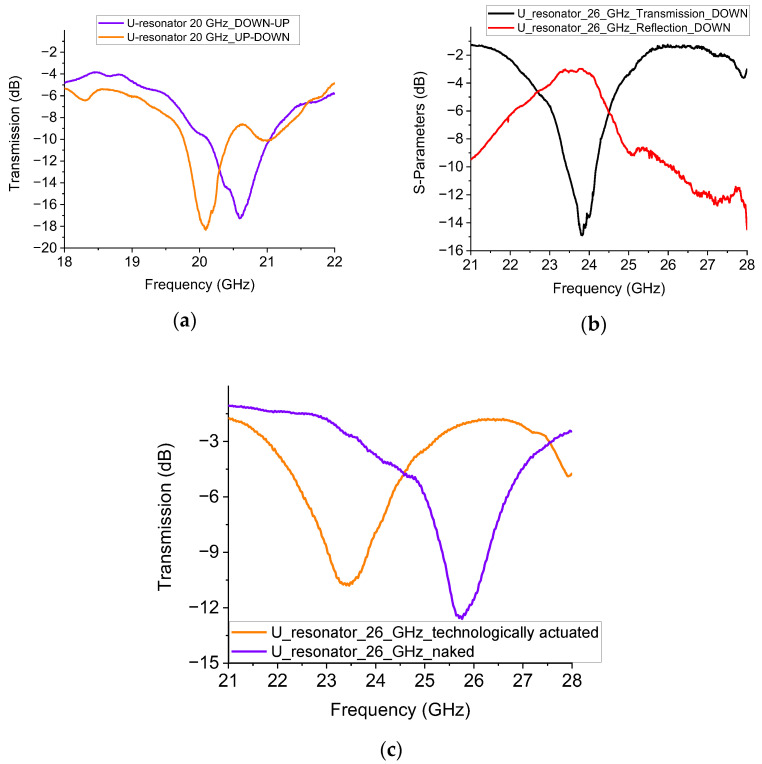
(**a**) The response of the 20 GHz semi-ideal device with technologically actuated switches and responses for the two situations (left or right arm switch actuated, corresponding to the up or down position of the metal beams). (**b**) The response of the 26 GHz resonator when the switch is technologically actuated. (**c**) The technologically actuated switch is compared with the naked one without a switch.

**Figure 8 sensors-23-00466-f008:**
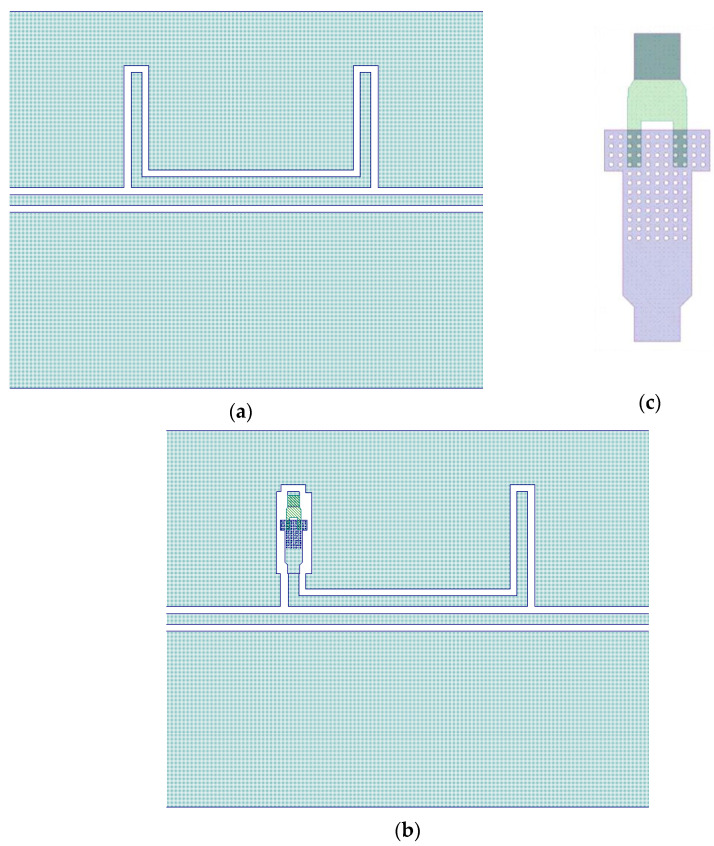
15 GHz, U-shaped naked resonator (**a**), the resonator loaded by the RF MEMS switch (**b**), and the detailed view of the cantilever switch (**c**).

**Figure 9 sensors-23-00466-f009:**
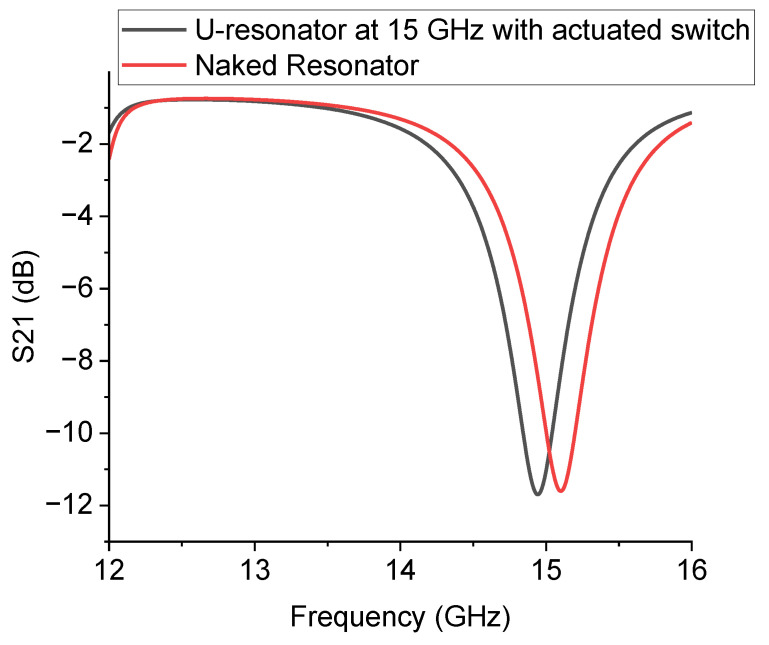
Simulation of the 15 GHz resonator with and without the presence of the RF MEMS switch. No effect is recorded on the matching and only a small shift in the resonance frequency is observed.

**Figure 10 sensors-23-00466-f010:**
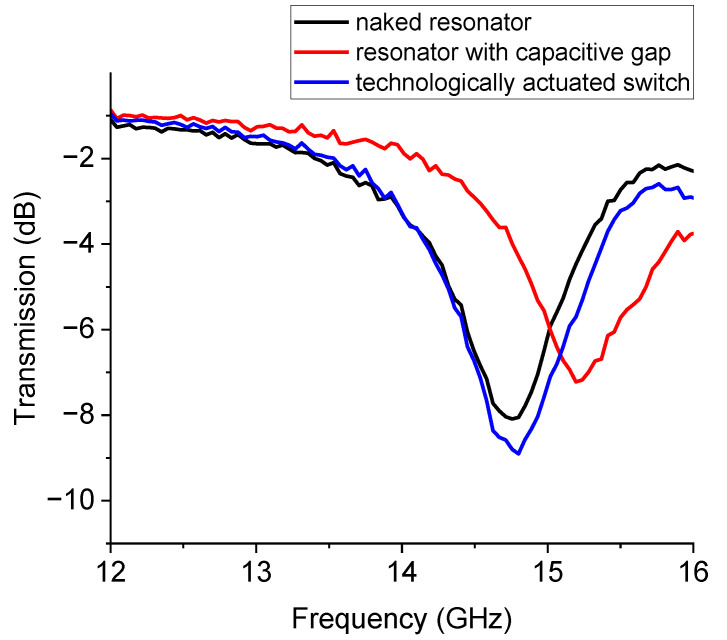
Experimental response of the U-shaped resonator for the 15 GHz operation. The black curve is for the U-resonator without a switch, and the blue curve is the response with the cantilever technologically actuated. The red curve is for the open switch, emulated by a capacitive gap.

**Figure 11 sensors-23-00466-f011:**
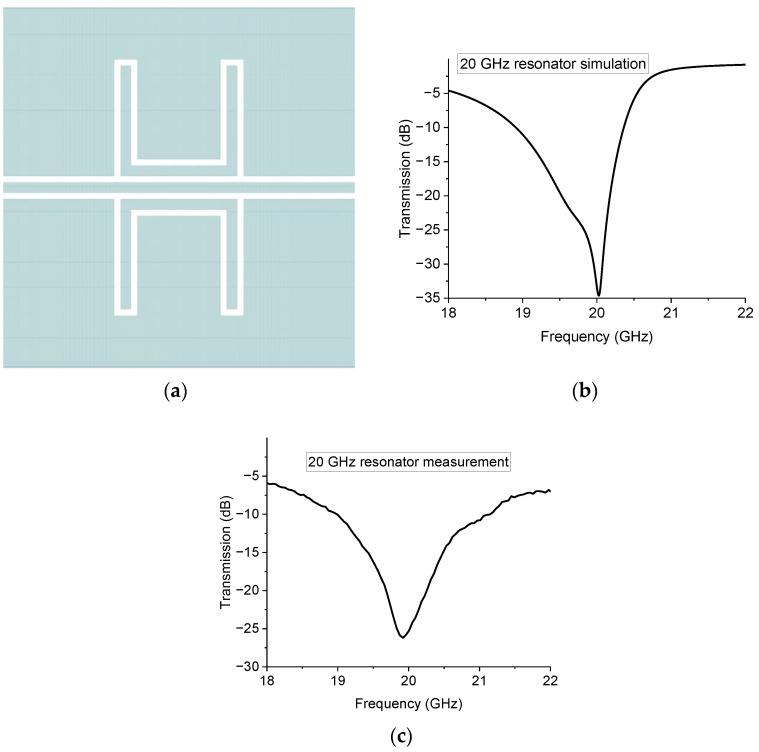
Symmetric implementation of the U-resonator for the 20 GHz. In (**a**) the resonator is mirrored on the other side of the central conductor of the CPW. Its predicted performance operation is shown in (**b**) and the experimental response is plotted in (**c**).

**Figure 12 sensors-23-00466-f012:**
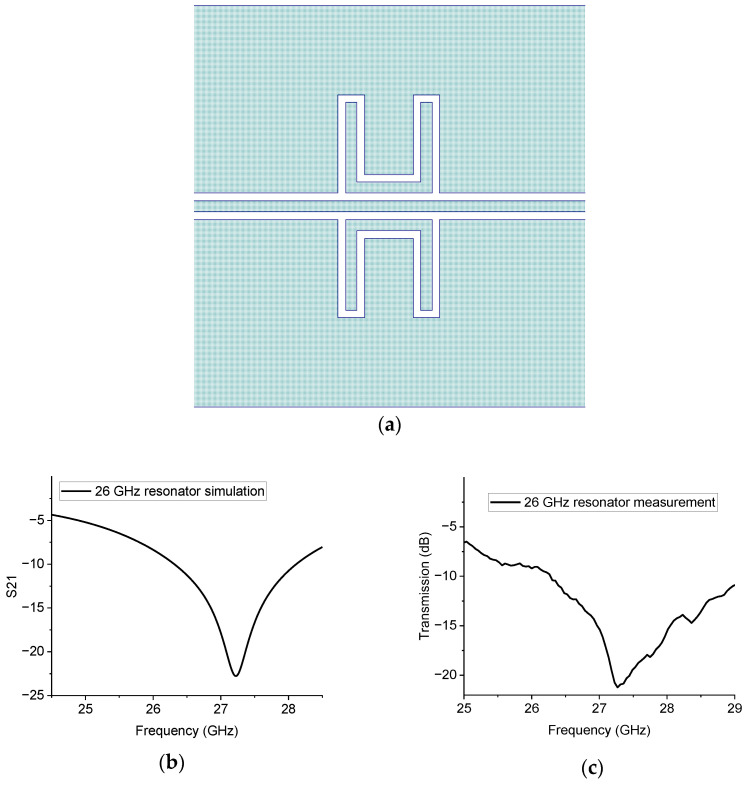
Symmetric 26 GHz resonator layout (**a**), theoretical expectation (**b**), and experimental measurement (**c**).

**Figure 13 sensors-23-00466-f013:**
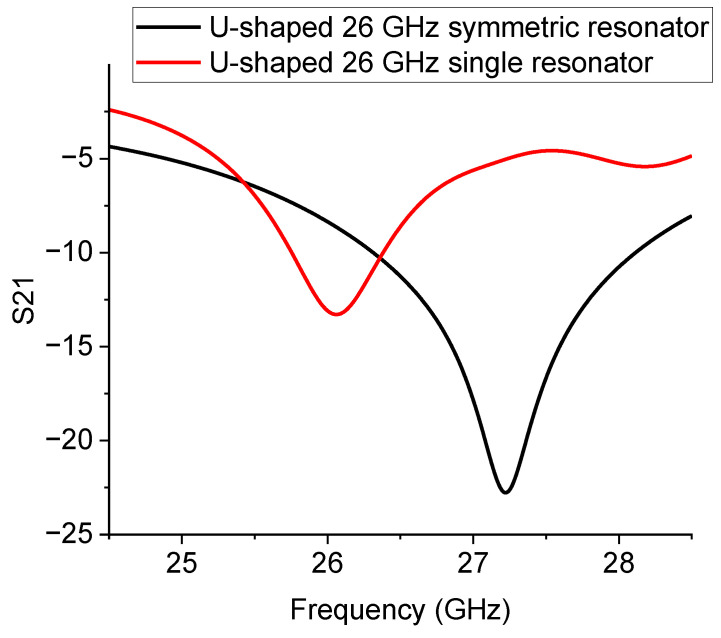
Comparison of the simulated 26 GHz resonators: the single and the mirrored configuration.

**Figure 14 sensors-23-00466-f014:**
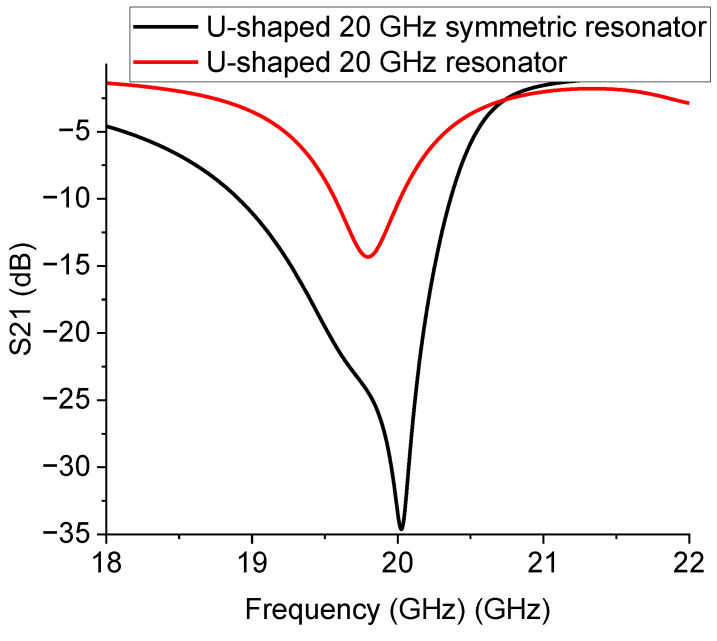
Comparison of the expected resonance responses for the 20 GHz single and mirrored resonators.

**Figure 15 sensors-23-00466-f015:**
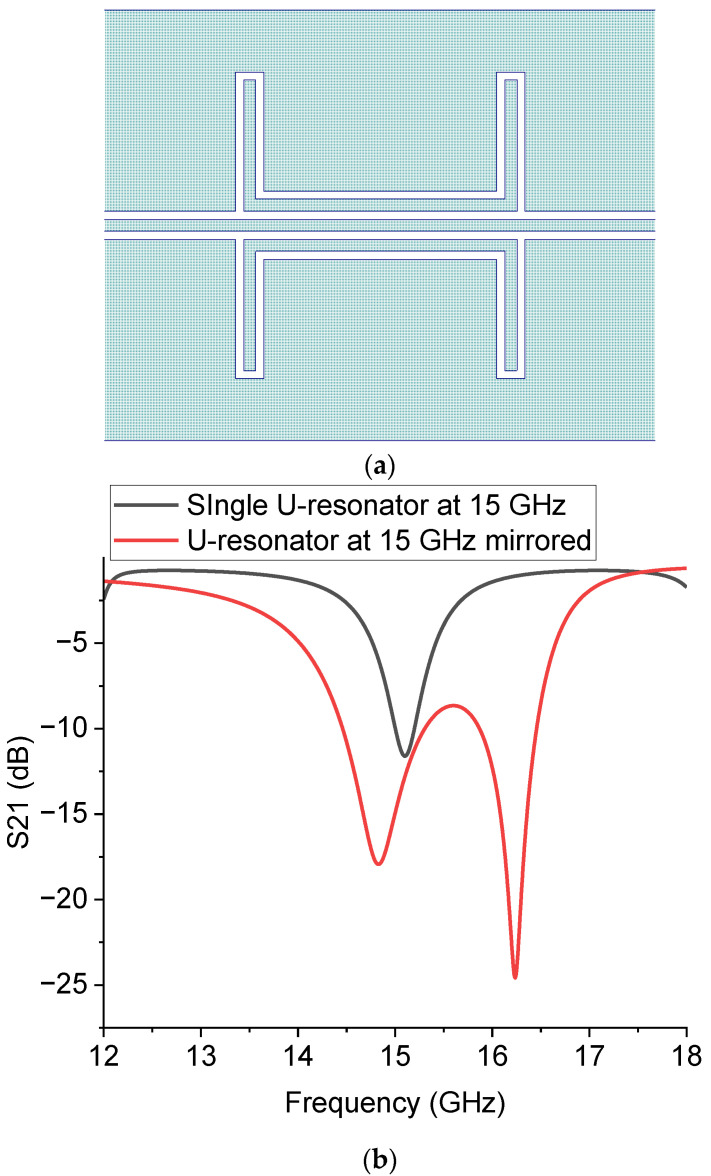
U-shaped resonator (**a**) and its predicted response comparing the single structure and the mirrored one (**b**).

**Table 1 sensors-23-00466-t001:** Sizes and geometrical details for the exploited U-shaped resonators.

Frequency [GHz]	A [µm]	L [µm]
15	980	810
20	725	1860
26	980	1950

## Data Availability

Raw data available upon request.

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
