# Peer review of "Design of U-Shaped Frequency Tunable Microwave Filters in MEMS Technology†"

_sensors, 2023, doi:10.3390/s23010466_

Round 1

Reviewer 1 Report

The conference extension of this paper is well written and could be recommended for publication, except for some minor revisions required.

1.     Please delete the text label in Line 67, Page 2.

2.     The bulk margin in Figure 1 is too large and could be pruned.

3.     The contents of the main text already presented at the conference could be shortened and compacted.

Author Response

  1. Please delete the text label in Line 67, Page 2.

Ok, we eliminated it as the paragraph is already focused on the device design, and we separated the technology explanation by numbering a purposely extended paragraph.

  1. The bulk margin in Figure 1 is too large and could be pruned.

Ok, we did it. Figure 1 has been resized to fit the available space within the text.

  1. The contents of the main text already presented at the conference could be shortened and compacted.

We reread the text explaining all the procedures to obtain the complete device, starting from the initial configurations manufactured to study the advantages and limitations of using RF MEMS switches. Sincerely, the contribution of the Conference (DTIP 2022) was limited to some device solutions. Therefore, we guess that repeating a full explanation of the step-by-step analysis of the devices could be more beneficial to the reader, avoiding combining information resulting from the activities of the same group. For this reason, we revised the text according to the Reviewer's suggestion, shortening the text a bit but without eliminating important considerations and figures helpful for immediate feedback on the performed research activities, including the proposed improvements.

Reviewer 2 Report

The manuscript looks good. The study of the U-shape resonator was described. However, in my opinion, the content is not that complete. The author should consider adding the following content:

1. Emphasise the novelty of the work in the introduction part.

2. Device fabrication process should be generally described. The authors have a reference in the main text. But still, general process should be described there.

3. The working mechanism and details of the cantilever switch is not mentioned at all. The authors should add some paragraphs to cover this part.

Author Response

in my opinion, the content is not that complete. The author should consider adding the following content:

  1. Emphasise the novelty of the work in the introduction part.

We included at the end of the introduction of our paper the following text to stress better the novelty of the work: " Despite few contributions in the literature about U-shaped resonators, no efforts are available, to the authors' knowledge, in the case of CPW-designed U-resonators implemented in MEMS technology. The novelty of our contribution is in the complete theoretical and experimental analysis that has been performed. This paper aimed to study: (i) the possibility of including micromechanical devices in the resonator structure accounting for the RF MEMS footprint, comparing double-clamped and cantilever switches, and (ii) the importance of symmetrical geometries to improve the electrical coupling of the resonator to the CPW feeding."

  1. Device fabrication process should be generally described. The authors have a reference in the main text. But still, general process should be described there.

Ok. We described the technology needed for producing the RF MEMS switches in more detail, and a numbered section was dedicated to the manufacturing process.

  1. The working mechanism and details of the cantilever switch is not mentioned at all. The authors should add some paragraphs to cover this part

We agree with the Referee. The following text has been included before the simulation of the cantilever-implemented resonators, just below the first period: " The cantilever-based switches adopt for the actuation a mechanism like that of a double-clamped beam used for the previous breadboards. In this case, the DC electrostatic actuation is provided by an actuation pad manufactured below the cantilever, fed by an external pad connected using a polysilicon highly resistive line. The actuation happens around the middle of the beam. This solution allows the collapse of the end of the cantilever resulting in a metal-to-metal contact located along the central conductor of the CPW. The same principle of operation is followed with the double-clamped structure, but we have two metal contacts instead of one only."

Reviewer 3 Report

I suggest the authors review the text edition to avoid a few inconsistent spacing after the sentence's endpoints

Author Response

I suggest the authors review the text edition to avoid a few inconsistent spacing after the sentence's endpoint

The text has been largely revised to check spacing and other typographical and grammar typos in the paper.